# The Inflammasomes Adaptor Protein PYCARD Is a Potential Pyroptosis Biomarker Related to Immune Response and Prognosis in Clear Cell Renal Cell Carcinoma

**DOI:** 10.3390/cancers14204992

**Published:** 2022-10-12

**Authors:** Jia-Qi Su, Xi Tian, Wen-Hao Xu, Aihetaimujiang Anwaier, Shi-Qi Ye, Shu-Xuan Zhu, Yue Wang, Jun Gu, Guo-Hai Shi, Yuan-Yuan Qu, Hai-Liang Zhang, Ding-Wei Ye

**Affiliations:** 1Department of Urology, Fudan University Shanghai Cancer Center, Shanghai 200032, China; 2Department of Oncology, Shanghai Medical College, Fudan University, Shanghai 200032, China; 3Shanghai Genitourinary Cancer Institute, Shanghai 200032, China; 4Department of Urology, The Affiliated Jiangsu Shengze Hospital of Nanjing Medical University, Suzhou 215228, China

**Keywords:** bioinformatics, renal cancer, microenvironment, biomarkers, immunology, PYCARD protein

## Abstract

**Simple Summary:**

Inflammation has been recognized as one of the hallmarks of cancers. PYCARD, the adaptor protein of inflammasomes, plays an important role in pyroptosis and apoptosis. However, the function of PYCARD remains unclear in human cancers. Here, we systematically performed a comprehensive analysis of PYCARD expression and its relationship with immunotherapy response and prognosis. We found significant differences in PYCARD expression between tumor and normal tissue, particularly in clear cell renal cell carcinoma. We also found that PYCARD was an unfavorable prognostic factor and was confirmed by external validation cohorts. Exploration of the profound mechanisms of PYCARD might help to identify new therapeutic targets and improve the efficacy of immunotherapy.

**Abstract:**

PYCARD is a protein engaged in inflammation, pyroptosis, and apoptosis. However, the function of PYCARD in human cancers remains unclear. The objective of our study was to explore PYCARD expression and prognostic value in human cancers. Public databases were used to assess PYCARD expression and prognostic value. The TISIDB database was used to explore the associations between PYCARD expression and different immune subtypes. The correlations between PYCARD expression and ICP genes, MMR genes, MSI, and TMB were also investigated. The immunotherapy response was assessed using the TIDE database. Single-cell RNA databases evaluated the PYCARD expression of immune cells. External datasets and immunohistochemical staining were conducted to validate PYCARD expression and prognostic value. The results showed that PYCARD expression varied in several cancers and was associated with prognosis, immune-related genes, published biomarkers, and immunotherapy response. Of note, PYCARD expression was upregulated in renal cancers with high diagnostic ability. Upregulation of PYCARD was correlated with worse prognosis in KIRC and external validation cohorts. In conclusion, PYCARD demonstrated strong correlations with prognosis, immune response, and disease progression in pan-cancer analysis. In ccRCC, PYCARD might serve as a biomarker for diagnosis and therapeutic target-boosting immunotherapy response.

## 1. Introduction

Inflammation, activated by extracellular molecules (pathogen-associated molecular patterns, PAMPs/danger-associated molecular patterns, DAMPs), is a classical biological response of human cells [1]. After activated by signals, cytokine cascade reactions cause spontaneous cell deaths [2,3,4]. The process of inflammation-related cell death is called pyroptosis [5]. Previous studies have developed pyroptosis-related signatures in evaluating immune cell infiltrations and prognosis [6,7]. The inflammation that causes pyroptosis has been considered a hallmark of cancer which has an impact on tumor progression [8]. The adaptor protein of inflammasomes, PYCARD, contains a caspase activation and recruitment domain (CARD) and a pyrin domain (PYD) [9,10,11]. Previous studies found that PYCARD had various functions in different human cancers [12,13,14]. However, the findings from different tumors were inconsistent and further investigations and validations are required.

PYCARD, known as apoptosis-associated speck-like protein containing a caspase recruitment domain (ASC), was firstly found to be related to apoptosis of human promyelocytic leukemia cancer cells treated with etoposide and all-trans retinoic acid [9]. PYCARD has also been involved in p53-Bax dependent apoptosis activated by caspase-2, caspase-3, and caspase-9 [15,16]. The disturbance of pro-apoptosis and anti-apoptosis homeostatic balance may correlate to carcinogenesis [17]. Tumor cells could escape from apoptosis by increasing anti-apoptotic signaling (Bcl-2 and Bcl-xL) and decreasing pro-apoptotic signaling (Bax and Bak) [18]. Indeed, it is generally accepted that PYCARD is a pro-apoptotic gene that suppresses tumor growth in many malignancies. In gastric cancer, PYCARD expression was higher in normal tissue than in tumors [13]. Similar conclusions were found in lung cancer [12], while in pancreatic adenocarcinoma, PYCARD expression was increased in 90% of tumor samples compared to adjacent normal tissue, and elevated PYCARD expression was associated with a poorer prognosis [14]. It is noteworthy that PYCARD exhibited pro-tumorigenic effects in pancreatic cancer, which suggested that PYCARD specks might communicate with infiltrating immune cells in the tumor microenvironment (TME) and eventually promote tumor progression rather than a pro-apoptotic effect [14]. However, due to the intricate cancer-specific genetic backgrounds, the significance of PYCARD in tumorigenesis requires systematic reassessments. 

In 2022, it is estimated that there will be 79,000 new patients diagnosed with renal tumors and 13,920 deaths in the United States [19]. Renal cell carcinoma is one of the most common types of malignant tumors. Clear cell renal cell carcinoma (ccRCC) accounts for around 75% of all renal tumors [20]. With the development of technology, the emergence of targeted therapy and immunotherapy improved the outcomes of advanced ccRCC patients. Immunotherapy has become the first-line treatment strategy for advanced RCC or patients with high risk [21]. However, the efficacy of immunotherapy still needs to be improved and the biomarkers distinguishing between responders and non-responders of immunotherapy are lacking. Thus, it is essential to explore the prognostic value and predictive ability of immunotherapy of PYCARD.

In our study, PYCARD expression and prognostic value were examined in the pan-cancer analysis. Furthermore, potential associations between PYCARD expression and immune subtypes, and published biomarkers were investigated. We next found that ccRCC had a higher level of PYCARD, which demonstrated strong correlations with clinical outcomes and immunotherapy response. Additionally, we confirmed PYCARD expression and prognostic value in ccRCC using external validation cohorts. Our study tried to reveal the underlying mechanisms of PYCARD in carcinogenesis and provide anti-tumor therapeutic insights into how to increase the efficacy of anti-tumor therapy.

## 2. Materials and Methods

### 2.1. Data Processing and Sample Collection

Transcriptome profiling (HTSeq-FPKM /HTSeq-Counts) and clinical characteristics of the Cancer Genome Atlas (TCGA) KIRC were obtained from UCSC Xena (https://xenabrowser.net/, accessed on 15 October 2021) [22]. The details of the Fudan University Shanghai Cancer Center (FUSCC) Proteomic Cohort were reported in the previous study [23]. The GEO datasets (https://www.ncbi.nlm.nih.gov/geo/, accessed on 15 October 2021) including GSE40435, GSE53757, and GSE36859, were downloaded to analyze PYCARD mRNA expression in ccRCC. The E-MTAB-1980 [24] and CheckMate-025 (CM-025) [25] were obtained and processed to analyze the survival differences between different PYCARD subgroups. Probes were averaged if the multiprobes were mapped to the same gene. Samples without survival information were excluded. The TCGA cohort contained 522 primary tumor tissues and 71 normal tissues, the FUSCC cohort contained 232 tumor tissues and paired 232 normal tissues, the E-MTAB-1980 cohort contained 101 patients, and the CM-025 cohort contained 311 patients. Previous paired samples of 26 patients with ccRCC from the Department of Urology at FUSCC (Shanghai, China) were collected during surgery and recruited for the studies. The Helsinki Declaration II was followed in the design of the study and the testing techniques. The study was approved by the ethics committee of Fudan University Shanghai Cancer Center (No. 2008222-Exp49, Shanghai, China).

### 2.2. PYCARD Expression, Genomic Alterations and Prognostic Value in Human Cancers

The Oncomine database collected published studies to investigate PYCARD expression differences in human cancers (accessed on 15 November 2021) [26]. The significant differentially expressed tumors were filtered by *p*-values < 0.0001 and Fold Changes > 1.5. The cBio Cancer Genomics Portal (c-BioPortal) (https://www.cbioportal.org/, accessed on 15 November 2021) was utilized to explore the genomic alteration of PYCARD among TCGA pan-cancer [27]. Pan-cancer PYCARD expression analysis was explored in TIMER (http://timer.cistrome.org/, accessed on 15 November 2021) [28]. The prognostic value of PYCARD expression was investigated using the GEPIA [29], Kaplan–Meier plotter (http://kmplot.com/analysis/, accessed on 15 November 2021) [30], and PrognoScan (http://dna00.bio.kyutech.ac.jp/PrognoScan/, accessed on 15 November 2021) [31] websites. The median values of PYCARD expression were used as cut-off values in GEPIA, and the cut-off values were determined automatically in the Kaplan–Meier Plotter database. Hazard ratios with 95% confidence intervals (CI) were determined. The pan-cancer survival analyses of PYCARD were obtained from PrognoScan database. 

### 2.3. PYCARD Expression Varied in Different Immune Subtypes 

TISIDB (http://cis.hku.hk/TISIDB/, accessed on 25 November 2021) database is a powerful online website for analyzing gene expressions and tumor immunity [32]. In our study, the TISIDB database was used to reveal relationships between PYCARD expression and immune subtypes in human cancers. 

### 2.4. Relationships between PYCARD Expression and Immune Checkpoint (ICP) Genes, Mis-Match Repair (MMR) Genes, Microsatellite Instability (MSI), Tumor Mutational Burden (TMB), and ESTIMATE Scores

The Pearson correlations between PYCARD and immune checkpoint genes, ESTIMATE scores [33], and tumor-related biomarkers (MMR, MSI, TMB) were investigated in pan-cancer analysis via Sangerbox website (http://www.sangerbox.com/tool, accessed on 25 November 2021), a powerful online website collecting and analyzing public datasets. ESTIMATE score analysis was performed to assess the stromal cells, immune cells in the tumor microenvironment (TME), and tumor purity [34].

### 2.5. PYCARD Expression and Survival Analysis in ccRCC

PYCARD expression from the TCGA cohort was explored using the “ggplot2” package [35]. The “Survminer” package was used to analyze outcomes. The diagnostic value and prognostic value of PYCARD were conducted using the “ROCR” and “TimeROC” packages [36]. External cohorts, including FUSCC Proteomic Cohort, GSE40435, GSE53757, and GSE36859 were utilized to validate PYCARD expression, and FUSCC Proteomic, E-MTAB-1980, and CM-025 cohorts were utilized to validate the PYCARD prognostic value. The best cutoff values were calculated using the “Survminer” package.

To further explore the PYCARD expression in advanced RCC (Stage IV) and early RCC (Stage I–III), we investigated PYCARD expression in different cohorts. The GSE40435 cohort was excluded because of a lack of stage information. The PYCARD expression comparisons were analyzed using the Wilcox test. Based on the above subgroups, we further detected the PYCARD prognostic value in advanced RCC or early RCC in three cohorts with survival information including TCGA, FUSCC Proteomic Cohort, and E-MTAB1980 cohorts.

### 2.6. Immunohistochemical (IHC) Staining Analysis

The tissue microarray (TMA) of clear cell renal cell carcinoma was purchased from Shanghai Zhuoli Biotechnology Co., Ltd. (Zhuoli Biotechnology Co., Shanghai, China). The TMA included sixty pairs of paired specimens incorporating pathological information. PYCARD protein expressions were detected using the Anti-PYCARD antibody (Abcam, ab283684, 1:400) according to procedures as previously described [37]. VisioPharm software was used to calculate the IHC signal. Using the analysis module, the region to be analyzed was segmented according to the staining intensity by applying the HDAB-DAB filter. Histochemistry score = ∑ (PI × I) = (percentage of cells with weak intensity × 1) + (percentage of cells with moderate intensity × 2) + percentage of cells with strong intensity × 3). PI refers to the proportion of the positive signal pixel area, and I refers to the coloring intensity.

### 2.7. Real-Time Quantitative PCR (RT-qPCR) Analysis

The total RNA of 26 patients was extracted using TRIzol reagent (Invitrogen Life Technologies, USA). The reverse transcription was performed using EZBioscience 4× EZscript Reverse Transcription Mix II (EZBioscience, Roseville, MN, USA). The Real-Time Quantitative PCR (RT-qPCR) experiment was conducted using EZBioscience 2× SYBR Green qPCR master mix (EZBioscience, USA) and detected using QuantStudio™ Real-Time PCR Software. All the experiments were conducted according to the manufacturer’s instructions. The primers for PYCARD were as follows: forward, 5′- TGG ATG CTC TGT ACG GGA AG-3′ and reverse, 5′- CCA GGC TGG TGT GAA ACT GAA-3′. The PYCARD expressions were calculated relative to that of GAPDH. Each sample was repeated three times and the average value was performed. The PYCARD mRNA expression was determined as 2^−ΔCt^ = 2^−(Ct (PYCARD) − Ct(GAPDH))^. The Wilcox test was used for the comparisons of the means of tumor and normal tissue.

### 2.8. Immunotherapy Response and Single-Cell Analysis of PYCARD 

TIDE (http://tide.dfci.harvard.edu/ accessed on 15 December 2021) is an online database that contains immunotherapy cohorts [38]. Liu2019-PD1-SKCM, VanAllen2015-CTLA-SKCM, and Braun2020-PD1-KIRC were used to detect PYCARD prognostic value in the immunotherapy cohorts. Next, TISCH (http://tisch.comp-genomics.org/, accessed on 15 December 2021) was used to evaluate single-cell PYCARD mRNA expression of immune infiltrating cells [39]. Four single-cell datasets, including ccRCC and melanoma, were selected (GSE139555; GSE145281; GSE148190; GSE120575, http://tisch.comp-genomics.org/, accessed on 15 December 2021). The Spearman correlations between PYCARD of different immune cell marker expressions were detected in the TIMER database. 

To eliminate the impacts of the confounding factors, we collected data from Checkmate clinical trial data to further investigate. JAVELIN signature, PBRM1 genomic status, MSKCC risk group, and IMDC risk group were obtained from the Checkmate clinical trial [25]. The anti-PD1 treatment cohort was selected for the following analysis. The best cutoff values were calculated using the “Survminer” package. The differences in efficacy of immunotherapy between different subgroups were identified using the Chi-square test. The PYCARD expression comparisons between different subgroups were analyzed using the Wilcox test. The survival analyses were performed using the log-rank method.

### 2.9. Analysis of Co-Expression and Protein–Protein Interaction (PPI) Networks

The linkedOmics database (http://www.linkedomics.org/login.php/, accessed on 15 January 2022) was utilized to establish co-expression networks in ccRCC through the Pearson correlation test [40]. Co-expression genes were used to perform Gene Ontology (GO) and KEGG pathways analyses. Affinity propagation with *p* < 0.05 was selected for the Biological Process (BP) and KEGG pathways using overrepresentation enrichment analysis (ORA) from the WebGestalt website [41]. GeneMANIA (http://genemania.org/, accessed on 25 January 2022) constructed PPI networks of PYCARD to investigate the molecules’ interacting networks [42]. The results from GeneMANIA were subsequently used for functional enrichment using the DAVID (https://david.ncifcrf.gov/, accessed on 25 January 2022) database [43]. The significant enrichment modules were filtered by false discovery rate (FDR) < 0.05.

### 2.10. PYCARD Subgroups Analysis

Patients from the KIRC cohort were classified into two groups based on the median expression of PYCARD (high vs. low). Differentially expressed genes (DEGs) between PYCARD subgroups were identified using the “DESeq2” package (|log2FoldChange| > 1 with *p*-value < 0.05). The “CIBERSORT” package with LM22 evaluated the 22 immune cells based on expression profiling [44]. The “GSVA” package was used to examine the single sample GSEA (ssGSEA) to explore the differences in 28 immune cell types between two subgroups [45]. Next, GO, KEGG, and gene set enrichment analysis (GSEA) were performed to investigate the different functional enrichments of two subgroups using the “ClusterProfiler” package (adjust *p*-value < 0.05 and FDR < 0.05). The GSEA analysis was conducted using “c5.bp.v7.0.symbols.gmt” gene set. Lastly, we explored the ICP gene expressions of two subgroups in KIRC cohort. 

### 2.11. PYCARD Expression and Drug Response

The pharmacological data downloaded from the CellMiner database were subsequently analyzed to explore the correlations between PYCARD expression and drug response using the Pearson correlation test in R software [46].

### 2.12. Statistical Analysis

The analysis was performed using R software (4.1.1, R Core Team, Vienna, Austria) and R packages. The figures were produced by Adobe Illustrator CC 2020 (Adobe Systems, San Jose, CA, USA). The distinctions between these two groups were analyzed using the Wilcoxon rank-sum test. All of the hypothetical tests had a significant *p*-value of 0.05 and were two-sided.

## 3. Results

### 3.1. PYCARD Expression, Genomic Alterations and Prognostic Ability in Human Cancers

The Oncomine database demonstrated that PYCARD was upregulated in bladder cancer, breast cancer, gastric cancer, head and neck cancer, leukemia, lymphoma, liver cancer, and other types of cancers. In contrast, PYCARD was downregulated in colorectal cancer, lung cancer, ovarian cancer, prostate cancer, and sarcoma. There were conflicting conclusions about PYCARD expression in kidney cancer (Figure 1A). The overall genetic alterations of PYCARD accounted for 1.1% of pan-cancer patients. The largest proportion of genomic alterations was amplification (Figure 1B). We next explored PYCARD expressions. The PYCARD was significantly higher in BLCA (bladder urothelial carcinoma), BRCA (breast invasive carcinoma), CESC (cervical squamous cell carcinoma and endocervical adenocarcinoma), CHOL (cholangiocarcinoma), ESCA (esophageal carcinoma), GBM (glioblastoma multiforme), HNSC (head and neck squamous cell carcinoma), KIRC (kidney renal clear cell carcinoma), KIRP (kidney renal papillary cell carcinoma), LIHC (liver hepatocellular carcinoma), STAD (stomach adenocarcinoma), and THCA (thyroid carcinoma) than in normal tissue. However, PYCARD was significantly lower in COAD (colon adenocarcinoma), KICH (kidney chromophobe), LUAD (lung adenocarcinoma), and PRAD (prostate adenocarcinoma) than in normal tissue (Figure 1C). The survival analysis demonstrated that a higher level of PYCARD was associated with worse OS (overall survival) in KIRC and LGG (brain lower-grade glioma), while a higher level of PYCARD was associated with better OS in BRCA, SARC (Sarcoma), and STAD (Figure 1D). As for DFS (disease-free survival), a higher level of PYCARD expression was only related to a prolonged DFS in UVM (uveal melanoma), while a lower level of PYCARD was correlated with better DFS in LGG (Figure 1E). The results obtained from the Kaplan–Meier plotter and PrognoScan are presented in Appendix A. The results above revealed potential links between tumorigenesis and PYCARD expression and suggested that PYCARD can be considered a potential prognostic biomarker.

### 3.2. PYCARD Expression Varied in Different Immune Subtypes

The immune landscape was previously conducted by Thorsson et al. [47]. Immune subtypes include six different subtypes which had unique immunological characteristics. The six immune subtypes, including C1 (wound healing), C2 (IFN-gamma dominant), C3 (inflammatory), C4 (lymphocyte depleted), C5 (immunologically quiet), and C6 (TGF-b dominant), were named by their gene expression signature. The C1 immune subtype was characterized by elevated angiogenic genes and a high proliferation rate. The C5 immune type had the lowest lymphocyte and the highest macrophages that were dominated by M2 macrophages. Our findings revealed the potential link between PYCARD expression and immune subtypes. Twenty types of cancer demonstrated significant correlations with PYCARD expressions, such as ACC, BLCA, SARC, and SKCM (skin cutaneous melanoma) (Figure 2). For example, PYCARD was downregulated in the C5 (immunologically quiet) group when compared with other immune subtypes in KIRC (Figure 2G), which indicated that PYCARD might play an important role in taking part in the crosstalk of the immune infiltrating cells within the tumor microenvironment (TME).

### 3.3. PYCARD Expression Correlated to Immune Checkpoint (ICP) Genes, Mis-Match Repair (MMR) Genes, Microsatellite Instability (MSI), Tumor Mutational Burden (TMB), and ESTIMATE Scores

Previous studies had found that ICP genes might impact immune cell infiltrations to affect the efficacy of the immunotherapy [48]. We found that ICP genes had significantly close correlations with PYCARD expression in almost pan-cancer (Figure 3A). The results concluded that PYCARD is significantly negatively correlated to MMR genes, especially in BLCA, BRCA, and CESC (Figure 3B). In contrast, MMR genes were positively correlated to PYCARD expression in LIHC. Microsatellite instability (MSI) was a tumor-specific trait correlated to mismatch repair (MMR) genes, which were proposed as biomarkers in colorectal cancer [49,50]. The results demonstrated that MSI had positive relationships with PYCARD expression in THCA, DLBC, and KIRC, and negative relationships in TGCT, SARC, READ, and LGG (*p* < 0.05) (Figure 3C). Tumor mutational burden (TMB) was identified as a biomarker authorized by the FDA for ICI treatment [51], and TMB was recognized as a biomarker in treating melanoma [51]. In previous clinical trials, better responses to immune checkpoint inhibitors (ICI) were found in patients with high TMB status, and TMB was found as a reliable marker for predicting immunotherapy response [52]. For TMB, PYCARD had negative relationships with TMB in THYM, TGCT, PRAD, LUAD, LIHC, LAML, COAD, and BRCA, and positive relationships with TMB in PAAD, LGG, GBM, and KIRC (*p* < 0.05) (Figure 3D). The relationships between PCYARD expression were conducted with ImmuneScore, StromalScore, and ESTIMATEScore (Appendix A). Except for CHOL, COAD, DLBC, ESCA, MESO, PAAD, READ, and UVM, the ImmuneScore of the other types of tumors showed significant associations with PYCARD expression. Among them, ACC, GBM, KICH, KIRC, LGG, SARC, and TGCT showed strong correlations with PYCARD expression (R > 0.6) (Appendix A). These observations above showed that PYCARD might have an immunological modulatory effect within the TME.

### 3.4. PYCARD Expression in ccRCC External Validation Cohorts

Next, we found that primary tumors demonstrated a higher level of PYCARD expression compared to normal tissue (Figure 4A). Survival analysis demonstrated that a higher level of PYCARD correlated with poor prognosis (*p* < 0.001) (Figure 4B), which was consistent with the above results. The area under curve (AUC) of PYCARD diagnostic value was 0.924 in the KIRC cohort (Figure 4C), and the AUC of PYCARD prognostic value was 0.640 at the first year, 0.636 at the second year, and 0.637 at the third year (Figure 4D). All four of the external validation cohorts demonstrated significantly higher levels of PYCARD in tumors compared with normal tissue (*p* < 0.05) (Figure 4E). The prognostic value of PYCARD was also confirmed by three external validation cohorts (FUSCC Proteomic Cohort, E-MTAB-1980, and CM-025) and we found that a higher level of PYCARD expression was an indicator of poor prognosis (*p* < 0.05) (Figure 4F). The details of the tumor microarray (TMA) of 60 paired ccRCC samples were presented in Appendix A. The immunochemistry staining of the TMA is presented in Figure 4G. After calculating the Histochemistry score, the results showed that PYCARD protein was higher in tumors than in normal tissues (*p* = 0.022) (Figure 4H). By applying qRT-PCR to 26 patients in the FUSCC cohort, we found that PYCARD mRNA expression was higher in tumors compared to normal tissues (*p* = 0.036) (Figure 4I). The results above confirmed that PYCARD was significantly higher in ccRCC and might play an important role in carcinogenesis.

Next, we further detected PYCARD expression in advanced RCC and early RCC. The results were contradictory. We found that PYCARD expression was significantly upregulated in advanced RCC compared to early RCC in TCGA cohort (*p* < 0.001) (Appendix A), while in other cohorts the differences disappeared (Appendix A). The results demonstrated that PYCARD could not serve as an indicator to evaluate whether the tumor metastasized or not currently. As PYCARD demonstrated significant prognostic value in external validation cohorts, we wondered if PYCARD expression could predict clinical outcomes in advanced or early RCC subgroups. The results of the subgroup survival analysis were statistically significant and highly variable (Appendix A). In TCGA, only in advanced RCC, PYCARD expression could provide extra prognostic value (*p* = 0.03) (Appendix A), while in the FUSCC Proteomic Cohort, PYCARD expression could provide extra prognostic value in early RCC (*p* = 0.02) (Appendix A). In the E-MTAB-1980 cohort, PYCARD could not provide extra prognostic value (Appendix A). The reasons for the discrepancy might be attributed to the sample size or other underlying factors, and the results from TCGA seemed more trustworthy because the cohort incorporated more samples compared to other cohorts. The PYCARD expressions between advanced RCC and early RCC and the prognostic value in stage subgroups should be validated in future studies.

### 3.5. PYCARD Expression Correlated to Immune Response

Biomarker analyses evaluated the predictive ability of PYCARD and published biomarkers for immunotherapy. The overall predictive value of PYCARD and other biomarkers is presented in Figure 5A. Then, we explored the PYCARD expression and outcomes in these cohorts. In SKCM, a high level of PYCARD correlated to prolonged OS and PFS (*p* < 0.05), while in KIRC, a higher level of PYCARD correlated to better OS and PFS (*p* > 0.05) (Figure 5B). The trend without statistical significance might be attributed to the limited number of patients and deserved further validation. Additionally, PYCARD demonstrated a comparable predictive ability compared with other predictive biomarkers in the Liu2019-PD1-SKCM, VanAllen2015-CTLA-SKCM, and Braun2020-PD1-KIRC cohorts (Figure 5C). Next, the PYCARD expression of different infiltrating immune cells was investigated at the single-cell level. The results revealed that PYCARD was mainly expressed on monocytes/macrophages and a small fraction of CD8+ T cells (Figure 5D,E). In addition, the correlations between TILs and cell marker RNA expression were investigated. The results showed that CD8+ T cells, T cells (general), and B cells had a strong relationship with PYCARD in KIRC (R > 0.4), and T cell exhaustion markers, including PDCD1 (R = 0.597), CTLA4 (R = 0.446), and LAG3 (R = 0.559) also demonstrated strong correlations with PYCARD, while this phenomenon was not found in SKCM (Table 1).

To further explore the underlying confounding factors, we utilized Checkmate clinical trial data to investigate the impacts of MSKCC stratifications, IMDC stratifications, PBRM1 status, and JAVELIN101 immune signature on immunotherapy response. The workflow is presented in Figure 6A. In the nivolumab treatment cohort, we firstly found that higher levels of PYCARD were correlated to OS and PFS (*p* < 0.05) (Figure 6B), which implied PYCARD predicting ability in evaluating prognosis. Next, we explored the ORR (objective response rate) in different subgroups. We found that only PBRM1 genomic status was associated with immunotherapy response (*p* = 0.012) (Figure 6C). PBRM1 mutation was associated with effective disease control in the anti-PD-1 therapy cohort, which was consistent with previous studies [25,53]. The other subgroups, such as JAVELIN signature, IMDC, or MSKCC did not demonstrate significant differences in ORR (Figure 6C). It was worth noting the trend that PYCARD and IMDC classifications were associated with ORR, although the differences were not statistically significant. We next explored whether these factors served as confounding factors on PYCARD as an immunotherapeutic biomarker. The PBRM1 status and JAVELIN101 immune signature demonstrated close relationships with PYCARD expression (Figure 6D), while the MSKCC and IMDC did not have this phenomenon. Patients with PBRM1 genomic mutations demonstrated a lower level of PYCARD compared to patients without mutations (*p* < 0.05). Additionally, patients with high JAVELIN signature seemed to have a higher level of PCYARD expression than patients with low JAVELIN signature (*p* < 0.001). Survival analysis showed that PBRM1 genomic status demonstrated significant associations with OS (*p* = 0.005), but not PFS (*p* = 0.05) (Figure 6E), while JAVELIN signature did not demonstrate significant associations with OS and PFS (*p* > 0.05) (Figure 6E). To confirm whether PYCARD has a predictive ability independent of these confounding factors, we again performed subgroups survival analysis based on the levels of confounding factors and PYCARD. In the PBRM1 mutation subgroup, patients with a low level of PYCARD expression were associated with better OS than patients with a high level of PYCARD expression (*p* = 0.02) (Figure 6F). Additionally, in the PBRM1 wild-type subgroup, there was also a similar trend (*p* = 0.08), which deserved further revalidations. In the JAVELIN low-signature subgroups, PYCARD expression could assist the JAVELIN signature to better predict the prognosis (*p* = 0.04) (Figure 6F). Similarly, the same trend was observed in JAVELIN high-signature subgroups (*p* = 0.06). It was also worth noting that once analyzed in PYCARD subgroups, the PBRM1 genomic status would not work as before (Figure 6F), indicating the complex regulatory networks in the ccRCC genomic background. We also found that MSKCC and IMDC classifications could predict OS (*p* < 0.001) but not PFS (Figure 6G). PYCARD could only assess prognosis in the MSKCC-favorable subgroup (*p* = 0.03) and there was a similar but not significant trend in the IMDC-favorable subgroup (*p* = 0.09) (Figure 6H). Future studies were needed for validating the conclusions. Our analysis explored the confounding factors and PYCARD correlations to immune response and demonstrated that PYCARD could provide extra value in predicting prognosis and treatment response. We revealed that PYCARD might take part in the immune regulatory system to impact the efficacy of the immunotherapy.

### 3.6. PYCARD Enriched Process of Immune Response, Inflammation and Apoptosis

The results above demonstrated that PYCARD expression was correlated with survival and immunotherapy response. The co-expression network was analyzed to find the underlying mechanisms. In KIRC, 7912 genes showed positive correlations to PYCARD and 5301 genes showed negative correlations to PYCARD (Figure 7A). The details of the co-expression genes are presented in Appendix A. The heatmap displayed the top fifty positive and negative related genes. (Figure 7B,C). Next, overrepresentation enrichment analysis (ORA) investigated the biological process of gene ontology (GO) analysis of PYCARD co-expression genes. The results were enriched in programmed cell death, cellular response to stress, protein transport, immune response, positive regulation of immune response, and cellular protein localization (Figure 7D). KEGG pathways demonstrated that its co-expression network mostly played a role in apoptosis, TNF signal pathways, and ribosomes (Figure 7E). GeneMANIA subsequently created a PPI network of PYCARD to explore its role in the process of tumorigenesis. PYCARD showed strong physical connections with AIM2, MEFV, PSTPIP1, NLRP3, PML, CASP1, NLRC4, and the PPI network, which were mostly involved in interleukin-1 related pathways (Figure 7F). The genes above analyzed in the DAVID database demonstrated that BP was correlated to positive regulation of interleukin-1 beta production, innate immune response, apoptotic process, apoptotic process, etc. (Figure 7G), while CC was correlated to several inflammasome complexes, IkappaB kinase complex, cytosol and cytoplasm (Figure 7H). Furthermore, MF was correlated to cysteine-type endopeptidase activator activity involved in the apoptotic process, identical protein binding, protein binding, and Pyrin domain binding (Figure 7I). The network of PYCARD was enriched mostly in the NOD-like receptor signaling pathway (Figure 7J).

### 3.7. PYCARD Expression Affected Immune Cell Infiltrations and Immune Regulation

Since PYCARD was differentially expressed in cancer and normal tissue, we performed differential gene analysis between high and low PYCARD groups. There were 1287 DEG in total, 369 downregulated genes and 918 upregulated genes shown in the volcano plot (Figure 8A). The proportion of immune cells was also investigated (Figure 8B). The CD8+ T, Treg, and gamma delta T cells had a higher proportion in the high PYCARD group. Conversely, monocytes, macrophages, CD4+ memory resting cells, and mast resting cells had a lower proportion in the high PYCARD group (Figure 8C). The majority of immune cells were all highly expressed in the high PYCARD group (Figure 8D). Following this, the DEGs were utilized to execute GO, KEGG, and GSEA investigations. The DEGs were enriched in T cell activation, epidermis development, etc. in BP; in the external side of plasma membrane, collagen-containing extracellular matrix, etc. in CC; and in the receptor ligand activity, signaling receptor activator activity, etc. in MF (Figure 8E). The DEGs were mainly enriched in combining interactions which played a vital role in cell signaling pathways and immune cell differentiation (Figure 8E). The GESA results were performed and presented in Appendix A. Enrichment results demonstrated that the high-PYCARD group was enriched in cytokine secretion, lymphocyte chemotaxis and migration, and response to tumor necrosis factor, while the low-PYCARD group was enriched in transmembrane transporter and transporter activity, plasma membrane region, and regulation of PH (Figure 8F). Lastly, twelve ICP genes were found to increase in the high-PYCARD group than in the low-PYCARD group in KIRC (Figure 8G). Our results undermined the potential influence of PYCARD in carcinogenesis.

### 3.8. PYCARD Expression Correlated with Drug Response

PYCARD expression was positively correlated with cyclophosphamide, HYPOTHEMYCIN, SB-590885, BGB-283, dabrafenib, GDC-0994, AZD-5991, AEW-541, ABT-199, hydroxyurea, MLN-2480, CT-GSK183, and ARQ-680. In addition, PYCARD expression was negatively correlated with MK-2461, BLU-667, and sonidegib. The correlation coefficient and P-value of the top 16 drugs are presented in Appendix A. The drug response analysis provided potential drugs in targeting PYCARD.

## 4. Discussion

PYCARD, the adaptor protein of inflammasome, plays an important role in tumor carcinogenesis and progression, as confirmed by increasing evidence [14,54]. In our study, we found that PYCARD expression varied among human cancers and correlated to prognosis. We further investigated the diagnostic and prognostic ability of PYCARD in ccRCC and verified its prognostic value in the FUSCC cohort. Moreover, we validated that PYCARD expression was correlated to immunotherapy response. The following functional enrichment analysis showed significant differences in immune response and transporter activity within the TME among two PYCARD subgroups. Therefore, our research indicated that PYCARD was a potential pyroptosis, apoptosis, and inflammation biomarker in predicting outcomes and immune response.

Pan-cancer analysis of PYCARD suggested that the function of PYCARD differed in different types of cancer due to distinct mutation backgrounds. However, the above results were not consistent with previous studies, which considered PYCARD as a suppressor gene in carcinogenesis [13,16]. In our study, only KIRC exhibited a stable negative correlation between PYCARD transcription and prognosis. Our results demonstrated that PYCARD had a close relationship with tumor-infiltrating lymphocytes and potential biomarkers. Enrichment analysis also showed that the high-PYCARD group was more likely to participate in chemotaxis and migration of lymphocytes. Although abundant infiltrations of infiltrating immune cells existed within the TME, patients with high PYCARD expression conversely demonstrated worse outcomes. The proportion of suppressive immune cells, such as MDSC and regulatory T cells, might be an explanation for the poor prognosis (Figure 8D). In addition, increased LAG3 and CTLA4 expressions in the high-PYCARD group suggested the presence of a suppressive immune microenvironment and exhaustions of active immune cells, which may promote immune escape and tumor progression [55,56]. In pancreatic adenocarcinoma, PYCARD secreted by tumor cells and tumor-associated macrophages (TAMs) stimulated thymic stromal lymphopoietin (TSLP) secretion of fibroblasts in the TME, and high PYCARD and TSLP expressions were linked to worse prognosis [14]. It could be the underlying crosstalk exists within the TME that restricts the integrity of the immune response, which needs to be further elucidated in future studies.

PYCARD expression was upregulated in the tumor samples and tumor cell lines of pancreatic ductal adenocarcinoma (PDAC) and PYCARD silencing contributed to the cell cycle arrest and decreased cell viability in PDAC [57]. Koizumi et al. attributed decreased cell viability to the decreased CCND1 expression which caused G1/S transition [57]. It is interesting to note that PYCARD silencing did not influence apoptosis-related expression levels in cancer cell lines. In glioma, similarly, ectopic PYCARD expression increased the viability and migration abilities of tumor cells [54]. These studies appeared to be independent of the pro-apoptotic effects of PYCARD. On contrary, Liu et al. showed that the PYCARD expression of renal cell carcinoma was downregulated compared to normal tissue, and tumor samples had a higher level of methylation than normal tissues [58], which was contradictory to our study. The differences might be attributed to selection bias. The contradictions between several studies might be due to the differences in patient choice and cancer genomic background. 

Another noteworthy point was the potential correlation of PYCARD with immunotherapy response. PBRM1 is the second most commonly mutated gene in ccRCC, correlates to immunotherapy response, and encodes a component of the SWI/SNF complex that regulates chromatin structure through ATP-dependent nucleosome remodeling [59]. The contradictory conclusions related to PBRM1 status on immunotherapy efficacy might be attributed to the patient selection, the clinical trial design, and anti-PD1 drugs [25,60], which requires future large clinical trials or multi-centers to validate the conclusion. Based on the Checkmate study, we found that PBRM1 status could serve as a biomarker in predicting immunotherapy response. We also found that PYCARD could provide extra prognostic value besides PBRM1 status and JAVELIN signature. Additionally, we found that PYCARD showed close relationships with PBRM1 status and JAVELIN signature [61]. The close associations might be attributed to the unique genomic background in ccRCC. We also found that some genes of JAVELIN 26 signature, such as PSIPIP1 and NLRC3, had interaction domains with the PYCARD protein. PYCARD could interact with PSIPIP1 and AIM2 (Figure 7F). Yang et al. found that PSTPIP1 could bind to CD2 to suppress T cell activation [62]. AIM2 was mainly expressed on Treg cells which were induced by TGFβ. Previous studies demonstrated that AIM2 could suppress glycolysis, but enhance the oxidative phosphorylation level of lipids, which might increase the stability of Treg cells [63,64]. Thus, we speculated that PYCARD might play an indispensable role in the formation of the suppressive immune microenvironment and eventually affect the immunotherapy response. 

Our study first symmetrically analyzed the role of PYCARD in the process of carcinogenesis and found the underlying mechanisms of the survival differences. Even though we conducted a complete and rigorous investigation of PYCARD and utilized various online tools and our FUSCC cohort for validation, this study has certain limitations. First, various databases differed and lacked specified analysis, potentially resulting in systemic bias. Second, in vitro experiments revealing the underlying mechanisms and unveiling the effects of the apoptosis protein on tumor-infiltrating cells in TME are needed. Third, the validation cohorts were limited, so future studies on the influence of PYCARD on immunotherapy should be further explored in large multi-centers to validate our results. Fourth, new techniques have been created to address heterogeneity, which may have been overlooked by bulk sequencing. Thus, future research should focus on elucidating PYCARD function in tumor growth with advanced technology.

## 5. Conclusions

In summary, PYCARD is significantly correlated to prognosis, immune response, and disease progression, suggesting that PYCARD serves as a potential indicator for prognostic value and immune response. PYCARD might be a therapeutic target and enhance the efficacy of immunotherapy. 

## Figures and Tables

**Figure 1 cancers-14-04992-f001:**
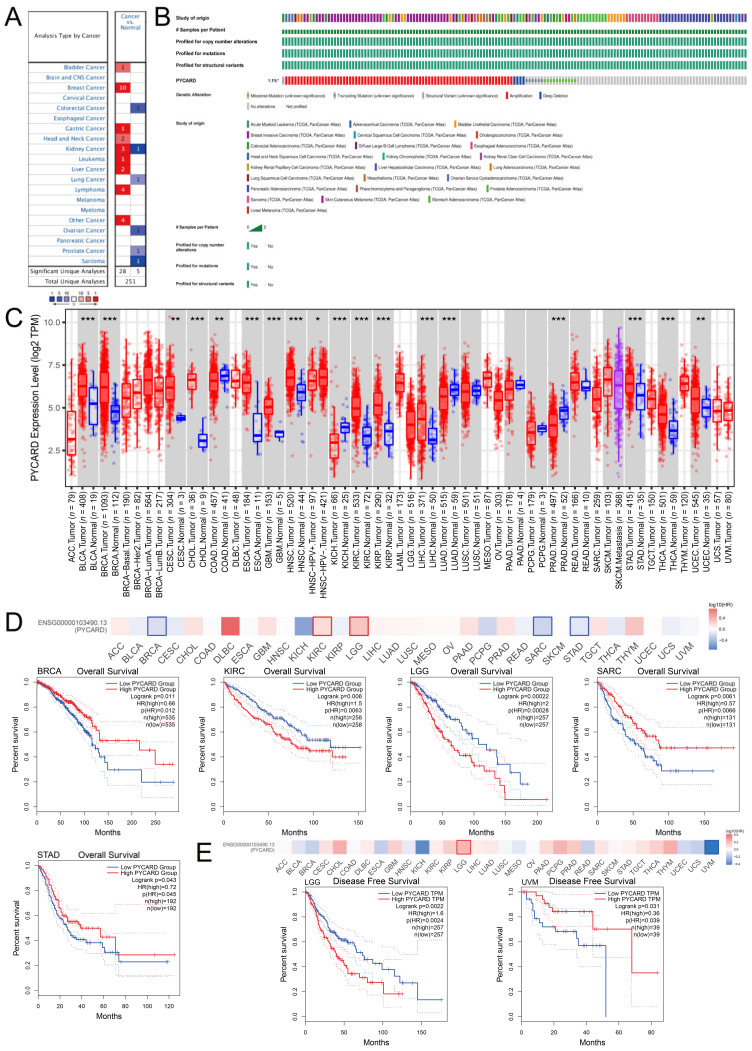
PYCARD expressions, genomic alterations and prognostic value in pan-cancer analysis. (**A**) PYCARD expression in pan-cancer analysis from Oncomine database. (**B**) The genomic alteration of PYCARD accounts for 1.1% of human cancers. (**C**) PYCARD expressions vary in different types of cancer. (**D**) Lower PYCARD expression is associated with worse OS in BRCA, SARC, STAD and higher PYCARD expression is associated with worse OS in KIRC and LGG (*p* < 0.05). (**E**) Lower PYCARD expression is associated with worse DFS in UVM and higher PYCARD expression is associated with worse DFS in LGG (*p* < 0.05). (* *p* < 0.05, ** *p* < 0.01, *** *p* < 0.001) (OS, overall survival; DFS, disease-free survival).

**Figure 2 cancers-14-04992-f002:**
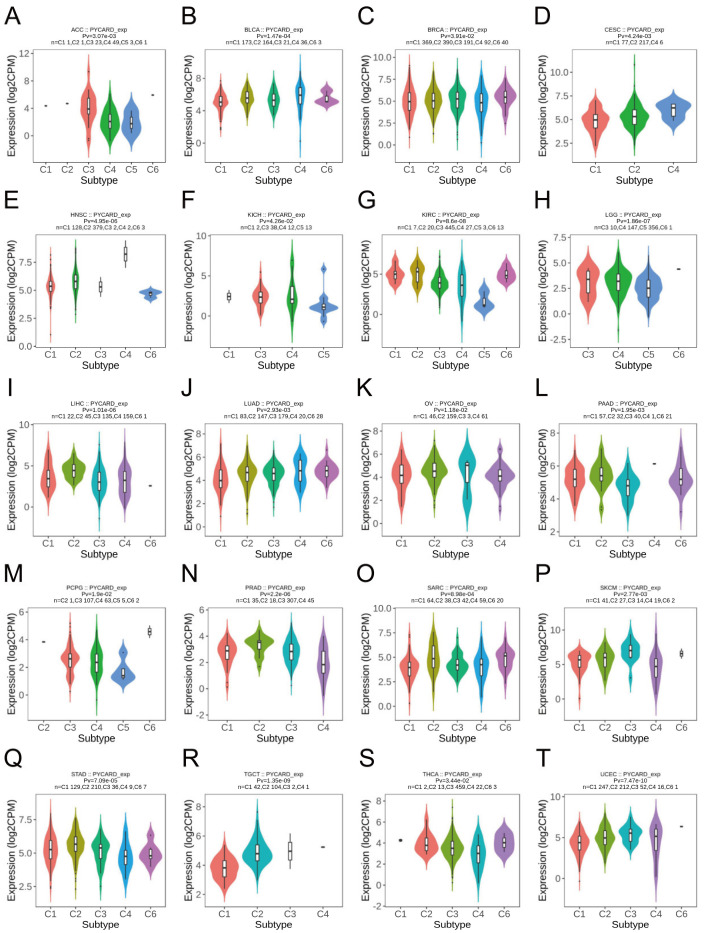
Associations between PYCARD expression and immune subtypes in human cancers. Twenty types of cancer including ACC (**A**), BLCA (**B**), BRCA (**C**), CESC (**D**), HNSC (**E**), KICH (**F**), KIRC (**G**), LGG (**H**), LIHC (**I**), LUAD (**J**), OV (**K**), PAAD (**L**), PCPG (**M**), PRAD (**N**), SARC (**O**), SKCM (**P**), STAD (**Q**), TGCT (**R**), THCA (**S**), and UCEC (**T**) demonstrate statistically significant associations between immune subtypes and PYCARD expression. *p* value was presented as Pv. (C1: wound healing, C2: IFN-gamma dominant, C3: inflammatory, C4: lymphocyte depleted, C5: immunologically quiet, C6: TGF-b dominant).

**Figure 3 cancers-14-04992-f003:**
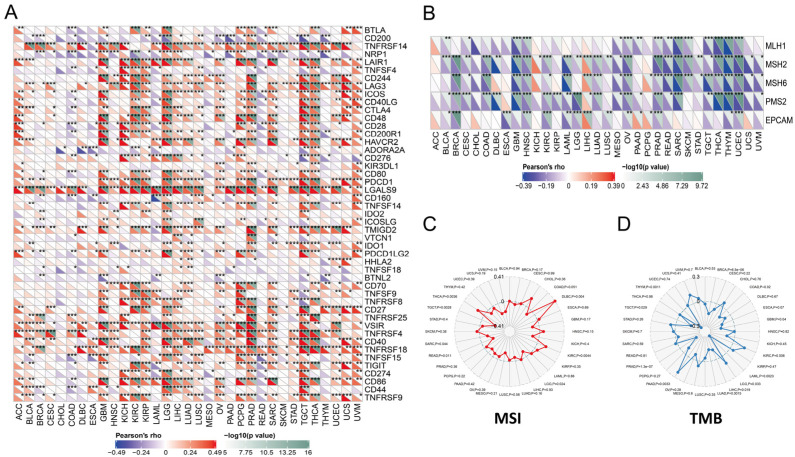
Pan-cancer associations between PYCARD expression and ICP genes, MMR genes, MSI, and TMB in human cancers. (**A**) PYCARD expression demonstrates significantly close associations with ICP genes in human cancers. (**B**) PYCARD expression demonstrates significantly close associations with MMR genes in human cancers. (**C**) PYCARD expression has positive relationships with MSI in THCA, DLBC, and KIRC, and negative relationships in TGCT, SARC, READ, and LGG. (**D**) PYCARD expression has negative relationships with TMB in THYM, TGCT, PRAD, LUAD, LIHC, LAML, COAD, and BRCA, and positive relationships with TMB in KIRC, PAAD, LGG, and GBM. (ICP, immune checkpoint genes; MMR, mismatch repair; MSI, microsatellite instability; TMB, tumor mutation burden) (* *p* < 0.05, ** *p* < 0.01, *** *p* < 0.001).

**Figure 4 cancers-14-04992-f004:**
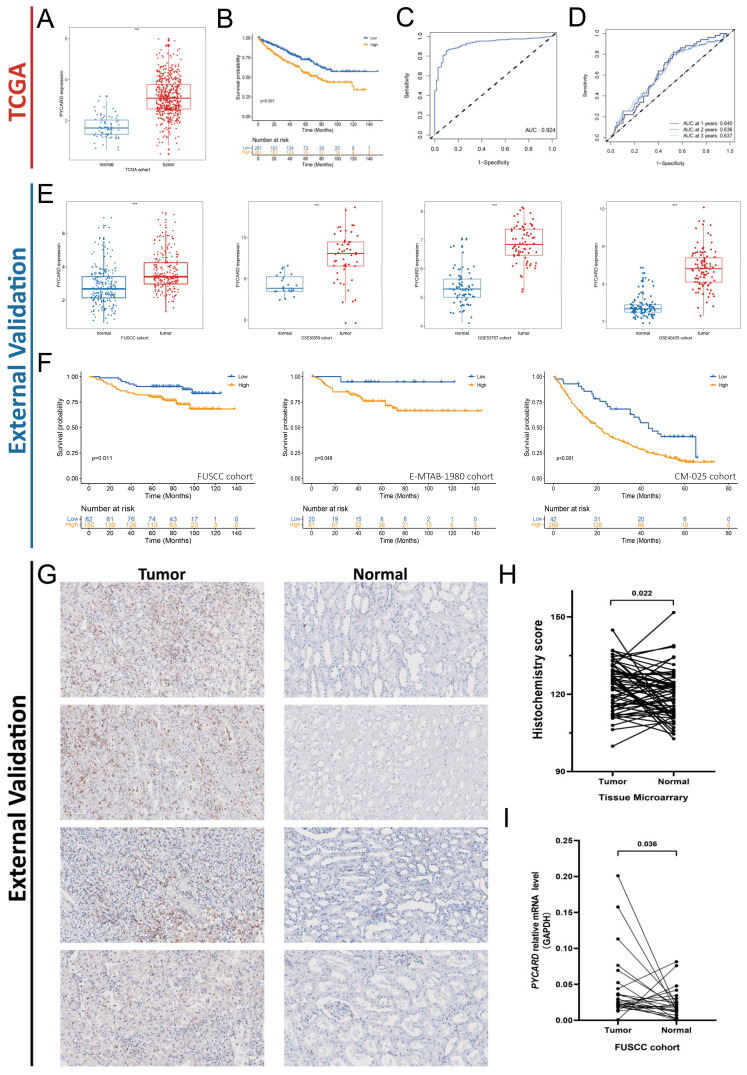
Upregulation of PYCARD expression is associated with prognosis in ccRCC. (**A**) PYCARD RNA expression is increased in ccRCC. (**B**) Higher PYCARD expression is associated with worse OS in KIRC. (*p* < 0.001) (**C**) Diagnostic value of PYCARD expression in KIRC cohort. (**D**) Prognostic value of PYCARD expression in KIRC cohort. (**E**) The PYCARD expression is upregulated in four external validation cohorts (FUSCC Proteomic Cohort, GSE40435, GSE53757, GSE36859). (**F**) Higher PYCARD expression is associated with worse OS in three external validation cohorts (FUSCC Proteomic Cohort, E-MTAB-1980, CM-025). (**G**) The immunochemistry staining of the TMA. The 20× imaging was used for scanning. (**H**) The quantitative histochemistry score of TMA demonstrated that PYCARD protein increased in ccRCC (*p* = 0.022). (**I**) Quantitative PYCARD relative mRNA expression of 26 ccRCC patients is increased in ccRCC compared to normal tissue (*p* = 0.036). (*** *p* < 0.001) (TMA, tissue microarray).

**Figure 5 cancers-14-04992-f005:**
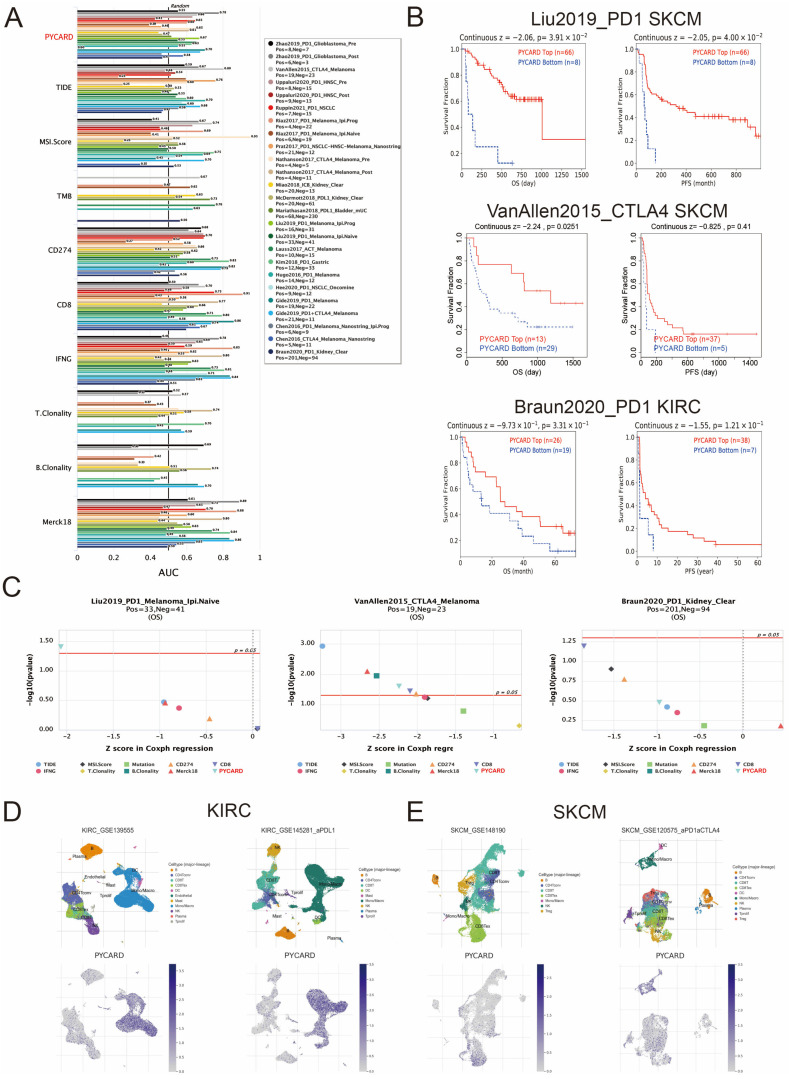
PYCARD expression correlated with immunotherapy response. (**A**) PYCARD predictive ability in immunotherapy cohorts. (**B**) PYCARD expression correlated with prognosis in SKCM and KIRC cohorts. (**C**) The comparison of PYCARD as a biomarker with other published biomarkers in SKCM and KIRC cohorts. (**D**,**E**) Single-cell analysis of PYCARD expression of immune cells in KIRC and SKCM cohorts.

**Figure 6 cancers-14-04992-f006:**
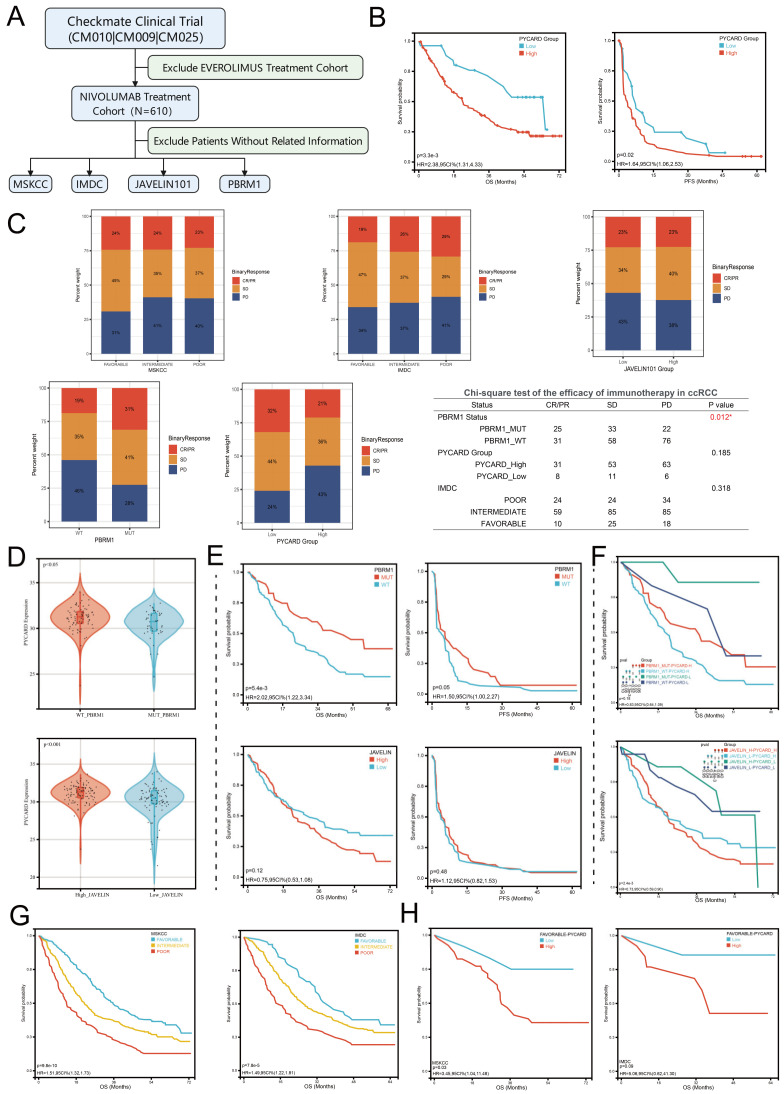
PYCARD expression is associated with immune response. (**A**) The workflow for analyzing associations between PYCARD and immunotherapy response. (**B**) Higher PYCARD expression is associated with OS (*p* = 0.003) and PFS (*p* = 0.02) (**C**) PBRM1 genomic status is associated with ORR (*p* = 0.012), while PYCARD expression demonstrated the trend but was not significant (*p* = 0.185). The Chi-square test was used for the analysis. (**D**) PBRM1 genomic status (*p* < 0.05) and JAVELIN signature (*p* < 0.001) are associated with PYCARD expression in the nivolumab treatment cohort. (**E**) Survival analysis of PBRM1 genomic status and JAVELIN signature in the nivolumab treatment cohort. Only PBRM1 status is significantly associated with OS (*p* = 0.0054). (**F**) Survival subgroup analysis of PYCARD plus PBRM1 genomic status or JAVELIN signature in nivolumab treatment cohort. (**G**) Survival analysis of MSKCC and IMDC subgroups in nivolumab treatment cohort. (**H**) MSKCC- and IMDC-favorable subgroups showed the trend that higher PYCARD expressions are associated with worse OS. (ORR, objective response rate).

**Figure 7 cancers-14-04992-f007:**
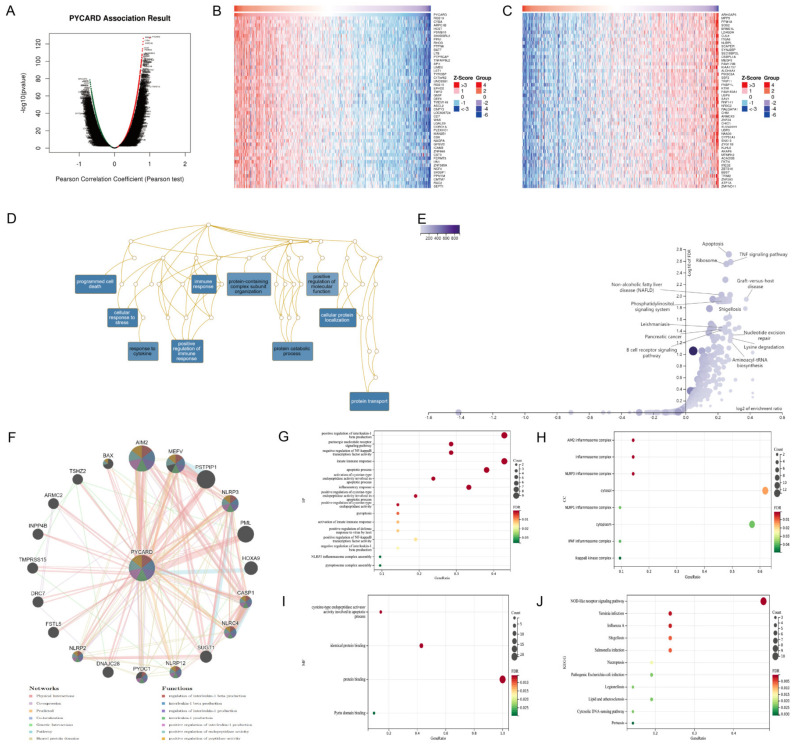
Functional enrichment of PYCARD co-expression and PPI networks. (**A**) Volcano plot of PYCARD highly correlated co-expression genes in KIRC. (**B**) Top 50 positive co-expression genes in KIRC. (**C**) Top 50 negative co-expression genes in KIRC. (**D**) The biological process of PYCARD co-expression genes enrichment in KIRC. (**E**) KEGG pathways of PYCARD co-expression genes enrichment in KIRC. (**F**) The PPI network of PYCARD and enriched pathways. Enrichment analyses of interaction genes were presented. Biological process in (**G**), cellular component in (**H**), molecular function in (**I**), and KEGG pathways in (**J**). (PPI, protein–protein interaction).

**Figure 8 cancers-14-04992-f008:**
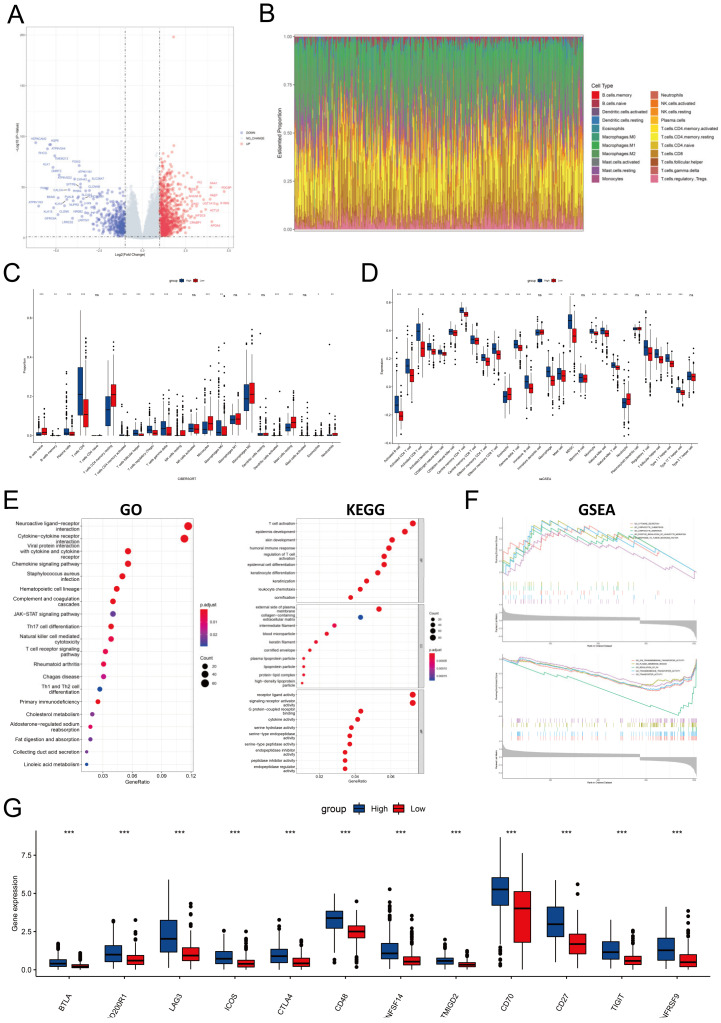
Bioinformatic analysis of two PYCARD subgroups in the KIRC cohort. (**A**) The DEGs between the high-PYCARD group and the low-PYCARD group are presented in the volcano plot. (**B**) The proportions of immune cells using the CIBERSORT method. (**C**) Difference of immune cell proportions between PYCARD subgroups. (**D**) Difference of single sample GSEA (ssGSEA) in immune cell proportions between PYCARD subgroups. (**E**) Gene ontology and KEGG pathway analysis of DEGs between PYCARD subgroups. (**F**) GSEA analysis of DEGs between PYCARD subgroups (**G**) Expressions of ICP genes varied in PYCARD subgroups (*** *p* < 0.001). (DEG, differentially expressed genes; GO, gene ontology; GSEA, gene set enrichment analysis; ICP, immune checkpoint).

**Table 1 cancers-14-04992-t001:** Associations between cell signature of immune infiltrating cells and PYCARD mRNA expression in KIRC and SKCM.

Immune Infiltrating Cell Markers		KIRC				SKCM			
		None		Purity adj.		None		Purity adj.	
Cell Type	Marker	Cor	*p*	Cor	*p*	*Cor*	*p*	*Cor*	*p*
CD8+ T cell	CD8A	**0.491**	*******	**0.422**	*******	0.224	*******	0.142	*******
	CD8B	**0.532**	*******	**0.472**	*******	0.242	*******	0.161	******
T cell (general)	CD2	**0.557**	*******	**0.490**	*******	0.222	*******	0.133	*******
	CD3D	**0.616**	*******	**0.559**	*******	0.247	*******	0.166	*******
	CD3E	**0.564**	*******	**0.500**	*******	0.259	*******	0.181	0.446
B cell	CD19	**0.468**	*******	**0.406**	*******	0.129	******	0.036	0.060
	CD79A	**0.493**	*******	**0.426**	*******	0.184	*******	0.088	0.318
Monocyte	CD86	**0.449**	*******	0.397	*******	0.068	0.140	−0.047	0.197
	CD115 (CSF1R)	**0.444**	*******	**0.401**	*******	0.047	0.308	−0.060	0.159
TAM	CCL2	0.014	0.741	-0.068	0.144	0.034	0.467	−0.066	0.315
	CD68	0.356	*******	0.332	*******	0.138	******	0.047	0.118
	IL10	0.287	*******	0.206	*******	0.018	0.691	−0.073	0.173
M1 Macrophage	INOS (NOS2)	−0.101	*****	−0.179	*******	−0.063	0.173	−0.064	******
	IRF5	0.383	*******	0.366	*******	0.198	*******	0.126	0.250
	CD80	0.337	*******	0.297	*******	0.049	0.286	−0.054	*******
	COX2 (PTGS2)	−0.133	******	−0.197	*******	−0.143	******	-0.169	*******
M2 Macrophage	CD163	0.203	*******	0.164	*******	−0.052	0.261	−0.169	*****
	VSIG4	**0.403**	*******	0.366	*******	−0.009	0.846	−0.094	******
	MS4A4	0.283	*******	0.219	*******	−0.020	0.661	−0.124	0.103
Neutrophils	CD66b (CEACAM8)	−0.095	*****	-0.096	*****	−0.086	0.062	−0.076	*****
	ITGAM	**0.426**	*******	0.392	*******	0.165	*******	0.103	******
	CCR7	0.399	*******	0.321	*******	0.234	*******	0.137	0.056
Natural killer cell	KIR2DL1	0.003	0.937	−0.029	0.530	0.130	******	0.090	0.081
	KIR2DL3	0.067	0.122	0.057	0.220	0.145	******	0.082	*******
	KIR2DL4	0.260	*******	0.223	*******	0.245	*******	0.183	*****
	KIR3DL1	−0.001	0.982	−0.022	0.631	0.169	*******	0.118	*****
	KIR3DL2	0.200	*******	0.166	*******	0.178	*******	0.103	0.052
	KIR3DL3	0.066	0.130	0.030	0.521	0.111	*****	0.091	0.233
	KIR2DS4	0.061	0.162	0.043	0.355	0.109	*****	0.056	0.233
Dentritic cell	HLA-DPB1	**0.502**	*******	**0.462**	*******	0.248	*******	0.056	******
	HLA-DQB1	0.313	*******	0.252	*******	0.235	*******	0.153	*****
	HLA-DRA	**0.410**	*******	0.361	*******	0.194	*******	0.103	******
	HLA-DPA1	0.392	*******	0.323	*******	0.216	*******	0.135	0.152
	HDAC1 (CD1C)	0.226	*******	0.142	******	0.156	*******	0.067	*******
	BDCA4 (NRP1)	−0.280	*******	−0.381	*******	−0.278	*******	−0.348	*******
	CD11c (ITGAX)	**0.422**	*******	**0.408**	*******	0.249	*******	0.180	******
Th1	T-bet (TBX21)	0.277	*******	0.209	*******	0.225	*******	0.145	0.783
	STAT4	0.310	*******	0.225	*******	0.111	*****	0.013	0.097
	STAT1	0.300	*******	0.230	*******	0.137	******	0.078	0.097
	IFN-γ (IFNG)	**0.496**	*******	**0.433**	*******	0.193	*******	0.078	*****
	TNF-α (TNF)	0.295	*******	0.250	*******	0.182	*******	0.098	*****
Th2	GATA3	0.241	*******	0.222	*******	0.206	*******	0.098	*******
	STAT6	−0.111	*****	−0.109	*****	0.155	*******	0.164	*******
	STAT5A	**0.424**	*******	0.360	*******	0.250	*******	0.245	*******
	IL13	0.033	0.442	−0.020	0.662	0.039	0.397	0.245	*******
Tfh	BCL6	−0.113	******	−0.139	******	−0.149	******	−0.183	0.569
	IL21	0.142	*******	0.120	******	0.101	*****	0.027	0.102
Th17	STAT3	−0.133	******	−0.206	*******	−0.061	0.185	−0.076	0.994
	IL17A	0.064	0.137	0.019	0.686	0.013	0.771	0.000	******
Treg	FOXP3	**0.544**	*******	**0.498**	*******	0.225	*******	0.143	0.668
	CCR8	0.370	*******	0.301	*******	0.076	0.099	−0.020	0.068
	STAT5B	−0.306	*******	−0.350	*******	0.083	0.072	0.085	0.448
	TGFB1	0.140	******	0.088	0.058	0.044	0.343	−0.036	*******
T cell exhaustion	PDCD1	**0.597**	*******	**0.552**	*******	0.295	*******	0.232	0.194
	CTLA4	**0.446**	*******	0.391	*******	0.134	******	0.061	*******
	LAG3	**0.559**	*******	**0.512**	*******	0.250	*******	0.177	0.902
	TIM-3 (HAVCR2)	0.131	******	0.075	0.107	0.104	*****	-0.006	*******
	GZMB	0.336	*******	0.271	*******	0.274	*******	0.209	*******

None, correlation without adjustment. Purity, correlation adjusted by purity. Cor. Value higher than 0.4 was considered as statistically significance and marked in bold. (* *p* < 0.05; ** *p* < 0.01; *** *p* < 0.001).

## Data Availability

The datasets used for the current study are available from online websites.

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
