# Peer review of "The Inflammasomes Adaptor Protein PYCARD Is a Potential Pyroptosis Biomarker Related to Immune Response and Prognosis in Clear Cell Renal Cell Carcinoma"

_cancers, 2022, doi:10.3390/cancers14204992_

Round 1
Reviewer 1 Report
Pyroptosis is a programmed cell death caused by inflammasomes, which is closely related to immune responses and tumor progression. The present study aimed to investigate the impact of PYCARD on Renal Clear Cell Carcinoma. PYCARD is significantly correlated to prognosis, immune response, and disease progression, suggesting that PYCARD serve as a potential indicator for prognostic value and immune response. PYCARD might be a therapeutic target and enhance the efficacy of immunotherapy. REFERENCES SHOUD BE ENRICHED AS FOLLOWS:
Lin G, Feng Q, Zhan F, Yang F, Niu Y, Li G. Generation and Analysis of Pyroptosis-Based and Immune-Based Signatures for Kidney Renal Clear Cell Carcinoma Patients, and Cell Experiment. Front Genet. 2022 Feb 24;13:809794
Sun Z, Jing C, Guo X, et al. Comprehensive Analysis of the Immune Infiltrates of Pyroptosis in Kidney Renal Clear Cell Carcinoma. Front Oncol. 2021;11:716854. Published 2021 Sep 9. doi:10.3389/fonc.2021.716854

Author Response
Comment 1: Pyroptosis is a programmed cell death caused by inflammasomes, which is closely related to immune responses and tumor progression. The present study aimed to investigate the impact of PYCARD on Renal Clear Cell Carcinoma. PYCARD is significantly correlated to prognosis, immune response, and disease progression, suggesting that PYCARD serve as a potential indicator for prognostic value and immune response. PYCARD might be a therapeutic target and enhance the efficacy of immunotherapy. REFERENCES SHOUD BE ENRICHED AS FOLLOWS:
Lin G, Feng Q, Zhan F, Yang F, Niu Y, Li G. Generation and Analysis of Pyroptosis-Based and Immune-Based Signatures for Kidney Renal Clear Cell Carcinoma Patients, and Cell Experiment. Front Genet. 2022 Feb 24;13:809794
Sun Z, Jing C, Guo X, et al. Comprehensive Analysis of the Immune Infiltrates of Pyroptosis in Kidney Renal Clear Cell Carcinoma. Front Oncol. 2021;11:716854. Published 2021 Sep 9. doi:10.3389/fonc.2021.716854
Response: We appreciate the reviewer’s kind recommendations. We have enriched our reference. Please find the corresponding revision on page 2, lines 50-51. Thanks so much again for your kind advice.
Reviewer 2 Report
Thank you for asking me to review this manuscript. In summary, the authors sought to understand the role of the PYCARD protein, particularly the prognostic role in clear cell RCC.
Despite the title and overall aim being related to RCC, most of the manuscript examines the role of PYCARD in a diverse mix of malignancies. If the authors want to focus on RCC, I suggest removing or streamlining much of the content on other tumor types and re-focusing the manuscript on RCC. They can include a summary of the data on the other tumor types within the background and discussion sections. In general, as an expert in RCC, I find the manuscript difficult to follow and the overall objective (association of PYCARD expression with prognosis in RCC) gets lost in overly convoluted details.
Other major comments:
- The authors quote - PYCARD was downregulated in the C5 (immuno-logically quiet) group when compared with other immune subtypes in KIRC. Was this statistically significant? Please provide more background on these different immune subtypes.
- The statement – PYCARD demonstrated an outstanding predictive ability compared with other predictive biomarkers in RCC is not justified. There are number of potential confonding variables that could influence the association between PYCARD expression and prognosis. The authors should collect clinical data on the IMDC prognostic group. They should also control for expression of PBRM1, which has been associated with improved prognosis in mRCC, specifically with 2L nivolumab in CheckMate 025.
- Can they comment on other prognostic gene expression signatures in RCC that have been published? Does PYCARD improve on these signatures? See 26 gene JAVELIN signature.
- Can the authors clarify if the discovery and validation cohorts contained both localized and metastatic RCC? If so, they should be done as separate analyses.
- Do the authors have data on impact of PYCARD expression on first-line IO treatment in mRCC?
- The results section reads like a methods section. Please edit the results section so that it summaries the data in RCC and not re-state the methods used to generate this data.
- Some of the figures are of low quality and it is difficult to read the fine print.
Author Response
Thank you for asking me to review this manuscript. In summary, the authors sought to understand the role of the PYCARD protein, particularly the prognostic role in clear cell RCC.
Comment 1: Despite the title and overall aim being related to RCC, most of the manuscript examines the role of PYCARD in a diverse mix of malignancies. If the authors want to focus on RCC, I suggest removing or streamlining much of the content on other tumor types and re-focusing the manuscript on RCC. They can include a summary of the data on the other tumor types within the background and discussion sections. In general, as an expert in RCC, I find the manuscript difficult to follow and the overall objective (association of PYCARD expression with prognosis in RCC) gets lost in overly convoluted details.
Response: We appreciate the reviewer’s kind recommendations and apologize for the convoluted results. In fact, we explored PYCARD expression and functions in pan-cancer at first. And then we found that PYCARD had a close relationship with immune system and immunotherapy-related biomarkers such as MMR, TMB and MSI. Lastly, we used external cohorts to validate PYCARD prognostic value and underlie potential links with immunotherapy response. To demonstrate the complete research route, we had retained all the pan-cancer analysis. However, it would lose the focus of our study. We apologize again for our unsatisfactory article structures. According to the reviewer’s suggestion, we selectively delete and simplify a portion of our pan-cancer analysis to focus on PYCARD that is a prognostic factor and in relation to the efficacy of immunotherapy. We also renew the introduction, discussion and figures. All changes are made in revision mode. Please find the corresponding revision of our manuscript.
Comment 2: The authors quote - PYCARD was downregulated in the C5 (immuno-logically quiet) group when compared with other immune subtypes in KIRC. Was this statistically significant? Please provide more background on these different immune subtypes.
Response: We thank the reviewer for this comment. The results presented in the new version of Figure 2 are all statistically significant. P value is presented as “Pv”. Please find the corresponding revision of Figure 2. The background of the six immune types was firstly developed by Thorsson on Immunity. We added the background of the immune landscape in our revised manuscript. Please find the corresponding revision on page 7, lines 266-273.
Comment 3: The statement – PYCARD demonstrated an outstanding predictive ability compared with other predictive biomarkers in RCC is not justified. There are number of potential confonding variables that could influence the association between PYCARD expression and prognosis. The authors should collect clinical data on the IMDC prognostic group. They should also control for expression of PBRM1, which has been associated with improved prognosis in mRCC, specifically with 2L nivolumab in CheckMate 025.
Comment 4: Can they comment on other prognostic gene expression signatures in RCC that have been published? Does PYCARD improve on these signatures? See 26 gene JAVELIN signature.
Response: We are grateful for the reviewer’s intuitive comments. We have re-chosen words to depict the predictive power of PYCARD as a biomarker. Please find the corresponding revision on page 11, lines 379 “comparable”. Thanks to the reviewer’s kind suggestion, our study re-investigated PYCARD associations with PBRM1 status, JAVELIN signature, MSKCC and IMDC classifications further. We obtained related information from the checkmate clinical trial and conducted the following investigations. Please find the corresponding revision on page 14, lines 399-439 and Figure 6. We compared MSKCC, IMDC, PBRM1 status and JAVELIN signature in the Checkmate nivolumab cohort and found that PBRM1 status significantly associated with immunotherapy response. While the JAVELIN 26 signature seemed not work in the Checkmate Study. PYCARD expression in specific group could provide extra prognostic value such as in PBRM1_MUT subgroup, higher PYCARD expression is associated with worse OS. We thank again the reviewers for their kind suggestions. We also found that PBRM1 mutation status and JAVELIN 26 signature showed close associations with PYCARD expression. The reasons were not fully investigated and should be studied in the future studies. Explanations based on our existing knowledge were presented. Please find the corresponding revision on page 20, lines 568-580. In addition, MSKCC and IMDC demonstrated overall survival stratifications but not PFS and ORR. PYCARD could also provide extra prognostic value in MSKCC or IMDC Favorable risk group. Our study revealed many details related to PYCARD expression and immunotherapy response; however, the conclusions need further validations by large cohort. What’s more, the underlying mechanism also need to be elucidated. We would improve the mechanism study in out next study.
Comment 5: Can the authors clarify if the discovery and validation cohorts contained both localized and metastatic RCC? If so, they should be done as separate analyses.
Response: We appreciate the reviewer’s kind recommendations. The GSE40435 cohort did not incorporate stage information. The rest of the cohorts including TCGA, GSE36895, GSE53757, E-MTAB-1980 and Checkmate had stage information to identify the patients whether the patient metastasized. However, here we only intended to validated PYCARD different expression between tumor and normal tissue and its prognostic value. As traditional prognostic biomarker research, survival analysis performed in the whole cohort was sufficient to draw the conclusion. However, based on the reviewer’s kind suggestions, we further explored PYCARD expression in advanced RCC patients (Stage 4) and early RCC patients (Stages 1-3). The results were conflicting that TCGA demonstrated that advanced RCC patients had a higher level of PYCARD expression than early RCC patients, while the other cohorts did not have this trend including Checkmate clinical trial. The differences of PYCARD in patients with metastasis or not should be re-validated in large cohorts by future studies. Our study tried to identified PYCARD expression between tumor and normal tissue and could not identify the PYCARD expression difference when metastasis occurred. The PYCARD expression difference between advanced RCC and early RCC needs further validations. Next, we explored PYCARD prognostic value in advanced and early RCC subgroups. Due to the Checkmate025 clinical trial patients were all advanced, the analysis was not incorporated this cohort. The results were hard to analyze. In TCGA, only in advanced RCC, PYCARD expression could provide extra prognostic value (P=0.03) (Figure S5G). While in FUSCC Proteomic Cohort, PYCARD expression could provide extra prognostic value in early RCC (P=0.02) (Figure S5H). In E-MTAB-1980 cohort, PYCARD could not provide extra prognostic value (Figure S5I). Please find the corresponding revision on page 10, lines 348-364 and Figure S5. Thus, without finding the key confounding factors, it is unsuitable to analyze in stage subgroups. The differences of PYCARD in patients with metastasis or not should be re-validated in large cohorts by future studies.
Comment 5: Do the authors have data on impact of PYCARD expression on first-line IO treatment in mRCC?
Response: We are grateful for the reviewer’s insightful comment. We have included that lack of large cohort validation into our limitations and we will improve it in our next study. Please find the corresponding revision on Page 20, Line 605-607.
Comment 6: The results section reads like a methods section. Please edit the results section so that it summaries the data in RCC and not re-state the methods used to generate this data.
Comment 7: Some of the figures are of low quality and it is difficult to read the fine print.
Response: We appreciate the reviewer’s comments. We have re-edited our manuscript and figures. Please find the new revision of our submission.
Reviewer 3 Report
The authors do an extensive bioinformatics approach using several databases and online tools, which sometimes gets a little confusing for the reader to follow.
I understand the importance of doing a pan-cancer analysis but since this manuscript is about ccRCC I think it would benefit from being more focused on ccRCC and the impact of PYCARD on this type of cancer.
Specific comments:
- figure 1 is blurred. Please increase the definition;
- lines 562-564 - this information regarding ccRCC epidemiology should be included in the introduction section;
- lines 576-585: the authors should also discuss that the size of their validation cohort should be increased to enable a stronger validation of the results obtained.
Author Response
Comment 1: The authors do an extensive bioinformatics approach using several databases and online tools, which sometimes gets a little confusing for the reader to follow.
I understand the importance of doing a pan-cancer analysis but since this manuscript is about ccRCC I think it would benefit from being more focused on ccRCC and the impact of PYCARD on this type of cancer.
Response: We appreciate the reviewer’s kind recommendations and apologize for the convoluted results. In fact, we explored PYCARD expression and functions in pan-cancer at first. And then we found that PYCARD had a close relationship with immune system and immunotherapy-related biomarkers such as MMR, TMB and MSI. Lastly, we used external cohorts to validate PYCARD prognostic value and underlie potential links with immunotherapy response. To demonstrate the complete research route, we had retained all the pan-cancer analysis. However, it would lose the focus of our study. We apologize again for our unsatisfactory article structures. According to the reviewer’s suggestion, we selectively delete and simplify a portion of our pan-cancer analysis to focus on PYCARD that is a prognostic factor and in relation to the efficacy of immunotherapy. We also renew the introduction, discussion and figures. All changes are made in revision mode. Please find the corresponding revision of our manuscript.
Comment 2: figure 1 is blurred. Please increase the definition.
Response: We thank the reviewer for this comment. As we have re-organized our study, the new version of Figure 1 was produced. Please find the corresponding revision of Figure 1.
Comment 3: lines 562-564 - this information regarding ccRCC epidemiology should be included in the introduction section;
Response: We are grateful for the reviewer’s helpful suggestion. The ccRCC epidemiology was re-organized and included in the introduction section. Please find the corresponding revision on Page 2, Line 76-85.
Comment 4: lines 576-585: the authors should also discuss that the size of their validation cohort should be increased to enable a stronger validation of the results obtained.
Response: We thank the reviewer for this comment. We have added the corresponding content in the discussion section. Please find the corresponding revision on Page 20, Line 594-597.
Round 2
Reviewer 2 Report
Thank you, no further comments.